# Green Consumer Behavior in the Cosmetics Market

**Nora Amberg** [1] **and Csaba Fogarassy** [2,*]

[1]  Doctoral School of Management and Business Administration, Szent Istvan University, Pater Karoly st. 1, 2100 Godollo, Hungary

[2]  Climate Change Economics Research Centre, Szent Istvan University, Pater Karoly 1, 2100 Godollo, Hungary

*  Correspondence: fogarassy.csaba@gtk.szie.hu

**Abstract:** Consumers and producers are becoming more open to the usage of natural cosmetics. This can be seen in them using a variety of natural cosmetic resources and materials. This fact is further supported by the trend of environmental and health awareness. These phenomena can be found within both the producers' and the consumers' behavior. Our research supports that green or natural products' role in the cosmetics industry is getting more and more pronounced. The role of science is to determine the variables suggesting the consumer to change to natural cosmetics. The primary aim of our research is to find out to what extent the characteristics of the consumption of organic foods and natural cosmetics differ. We would like to know what factors influence consumer groups when buying green products. The novelty of the analyses is mainly that consumers were ordered into clusters, based on consuming bio-foodstuffs and preferring natural cosmetics. The cluster analysis has multiple variables, namely: Consumer behavior in light of bio-product, new natural cosmetics brand, or health- and environmental awareness preferences. The data was collected using online questionnaire, exclusively in Hungary during April–May of 2018. 197 participants answered our questions. The results of descriptive statistics and the cluster analysis show that there are consumers who prefer natural cosmetics, whereas some of them buy traditional ones. A third group use both natural and ordinary cosmetics. The results suggest that on the market of cosmetic products, health and environmental awareness will be a significant trend for both producer and consumer behavior, even in the future. However, it will not necessarily follow the trends of the foodstuffs industry, as the health effect spectrum of cosmetics is far shorter. In the future, the palette of natural cosmetics will become much wider. The main reason for this will be the appearance of green cosmetics materials and environmentally friendly production methods (mostly for packaging). The consumers will also have the possibility to choose the ones that suit them the most.

**Keywords:** natural cosmetics; green cosmetics; bio-foodstuffs; consumer behavior; consumer preference

## 1. Introduction

Currently, a variety of greentech solutions are available in the cosmetics industry. Using these solutions, environment-friendly natural cosmetics can be produced. There is a multitude of research into the use of new environmentally friendly technological solutions as well. This study mainly focuses on introducing the effects of environmental and health awareness trends on the cosmetics industry, for both the producer and consumer sides. The main point of the analysis was to perfectly identify the factors influencing green consumer behavior. The factors important for the analysis of research questions were found using primary research, specifically, online questionnaire. Based on the literature sources, the following were reviewed: A short history of natural cosmetics, the specifics differentiating between cosmetics, the development of environmental protection principles, the security regulations

of the European Union related to cosmetics, the producer/organisational behavior for producing cosmetics, and the consumer behavior related to cosmetics. Our research was partly based on the 2016 research of Matić and Puh [1]. They already conducted a similar analysis for the Croatian consumers. However, they used a different method for their analysis (binary logistical regression model). They also didn't take note of the connections with foodstuffs consuming habits. In light of these points, our research wants to find first and foremost, the effects of the trends of consuming bio-foodstuffs and using natural cosmetics have on each other. Furthermore, we wish to determine grouping for consumers based on their usage of natural cosmetics, and find possible future trends based on the groups formed. The main purpose of our research is to gain a clear picture of the similarities and differences in the consumption habits of organic foods and natural cosmetics. The results of these studies allow us to draw conclusions about what additional information is needed to develop healthy consuming culture. By learning about each group, communication and education strategies or programs can be defined.

## 1.1. A Short History of Cosmetics

The history of cosmetics can be said to have started during ancient Egypt. Their usage had, first and foremost, hygienic purposes and health advantages [2–4]. The usage that also has advantages for healthcare, or fighting against the aging of the skin are relatively new approaches to cosmetics. The 'cosmeceuticals' (which is a combination of "cosmetics" and "pharmaceuticals") word was first used by Albert Kligman in 1984, in order to have an expert definition of products offering both cosmetic and therapy value. Beauty, however, comes from the inside. The aging of skin also has a significant relation with eating habits—which is also supported by multiple instances of research [2,5]. These research materials also describe the skin's history, its chemical constitution, and sources. Researchers were able to prove that a variety of materials have a positive effect on dermal health either in their effect mechanism, or in their function. They used mainly animal and clinical research and experiment data to determine biological and bio-medicinal effects for the following: Collagen, Ceramide, Beta-carotid, Astaxanthin, Coenzyme $Q_{10}$, colostrum, Zinc, and Selene. During the Antiquity Age, people used natural elements (water, salts, and metals) and plant and animal extracts for healthcare and cosmetic goals [2,6,7].

Nowadays, the trend of using and seeking natural materials and additives is on the rise. This is most notable for cosmetic products. The reason for the increase in popularity is that the negative effects synthetic materials have on health and the environment were made apparent. Currently, marketing trends are turning towards natural solutions for cosmetics, which have a relation to healthy lifestyle, and link cosmetic product usage to healthy eating habits [6]. During recent decades, the role of cosmetics degraded on the level of the entirety of society. Many products became a natural part of everyday life. Most products, like soap, shampoo, toothpaste, and such, became parts of our everyday hygiene, where most notable preferences are based on price level, instead of the environmental-friendliness of the product. A similar effect could be observed for sunscreens, where the protection of the skin is mandatory. Makeup is also a natural part of everyday life now, they are a tool for a confident look [8]. Based on our ethno-botanical knowledge, we can state that consumers used natural materials for skincare and to improve their looks until recently. However, due to healthcare problems becoming global, and interest in skincare (mainly due to UV rays) becoming more intensive, the need for much more efficient plant extracts became more pronounced [9]. Therefore, we can assume that in the future, the trends of eating habits and cosmetics may separate, and show a different tendency in the future.

The history of cosmetics is shaped parallel to that of humanity, which had relied on fishing, hunting, and superstitions in its early days. Later, it turned towards medicine and pharmacies. Cosmetics also changed along with the changes in humanity. Nowadays, the cosmetics market is vastly different from the industry mentioned earlier. It became incredibly competitive and global, where quality, efficiency, and safety are all highly important. Consumers also became extremely refined, therefore, scientific research and product development became steps producers cannot skip.

Furthermore, consumers are well aware of environmental protection and sustainability questions (animal protection, active agent—pollutant relations). Therefore, new cosmetic ingredients also have to pass an environmental protection criterion. The newest development trends of cosmetics are based on researching the natural ingredients that block skin aging.

*1.2. Main Specifics of Cosmetic Products, Practice of Using Natural Materials*

Current environmental problems are stimuli for the consumers, encouraging them to buy green products. Green or natural products are developed along ecological standards, and perfected as such. Green products have a variety of advantages, for example: Less water, material and energy usage during production, non or slightly pollutant to natural environments, and their packaging can be recycled [10,11]. Commercially available green products also include green cosmetics. Consumers devote more and more time to understand these products. Green or natural cosmetics are made out of natural resources, without the usage of chemicals, coloring additives, or other non-natural mixtures [12,13]. Green cosmetics are also often called organic cosmetics, which should not be mistaken for each other. Organic cosmetics have a much more strict definition, and selling them in consumer systems can also be a significant challenge (storage, expiration, etc.). The reason is that organic cosmetics have to offer a maximized environmental efficiency, stability, and security [11]. Green cosmetics are often more expensive, which may result in less consumers being able to buy them. Interest in green, sustainable, and natural products, however, is on the rise on the market of cosmetics and body care products [14]. Green cosmetics are multi-faceted constructions usually aimed at the following: Environmental conservation, minimisation of polluting, responsible usage of non-renewable resources, and preservation of fauna and species. Green or environmentally friendly products are mainly products defined as non-hazardous to natural resources and renewable. They can basically be used without harming the environment. Green cosmetics are natural cosmetics, primarily made up of plant and fruit extracts and concentrates [15].

According to the documentation on plants used as cosmetics "Choa Arabian and Kotoko ethnic groups' knowledge in Korusseria (Far North Region of Cameroon)", trees are the plants most used for cosmetic purposes, mainly by using their bark and seeds. People in this region use more than 40% of the registered plants found here for skincare. The valuable phytochemical ingredients in cosmetics can be found in all of said plants. These plants can also be used effectively in dermatology, furthermore, they have an anti-oxidant property. They can serve as materials for perfume, have anti-inflammatory, anti-microbial, and curative effects on wounds. They have a skin whitening effect as well, can be applied to tooth cavities, they constrict skin pores and gums, and are effective materials in hair care. Usage of the plant cosmetic ingredients and the ancient knowledge of the Choa Arabian and Kotoko people are tightly linked, however, most of these plants are less researched in the industry of cosmetics [16]. For example, more and more cosmetic products contain shea butter. Shea trees grow on 4 million square kilometres in Sub-Sahara Africa. Shea parks are a sustainable source for consumable fats (shea butter) [17].

Ingredients that come from marine creatures and algae are also found in many cosmetics [14]. Many biologically active materials that can be used for cosmetic purposes can be extracted from marine lifeforms. Such as algae (macro- and micro-algae) as a prime example. Their usage in cosmetic products can be accredited to their skincare attributes. Algae hydrate the skin, aid circulation, activate the renewal and metabolism of cells, regulate the operation of sebaceous glands, regenerate tissue, have an anti-inflammatory effect, and increase skin resistance [18].

Bio-active ingredients coming from natural sources have a well-known positive effect in cosmetic usage, which also serve as incentive for consumers. Of these, gallotannins have a very intriguing potential. Caffeic acid (CAF) is one of the most promising active ingredients, since it is an anti-oxidant, anti-inflammatory, and anti-wrinkle as well. In case of local usage, increasing its biological availability may lead to source material expecting new cosmetic interest. Clay minerals also have exceptional qualities, among others, low or no toxicity, and high bio-compatibility [19].

Color is a key factor in the product, which may be an incentive to consumers. Cosmetics can have added coloring in order to color either the product, or a part of the body (skin, hair, nails, and eyelashes). In case of the latter, color cosmetics is the area showing a high-level growth within the cosmetics industry. There are cosmetics which are applied to the skin for a longer time, like rouge, creme, and body lotion. There are also those that can be washed off, removable shortly after use, like shampoo, gel, and soap [20,21].

Colorings can be classified by their structure, source, color, method of application, and solubility. According to solubility, there are two categories: Colorings and pigments. Colorings are synthetic organic materials which are soluble in water or oil within the cosmetic product, like skincare or finery products. Pigments are not soluble, they remain inside the molecules, therefore, they are mainly used in toothpaste, or decorative makeup products [20–22]. Among the thousands of materials used as coloring, synthetic coloring is more advantageous to produce compared to natural (extracted from plants, animals, and minerals). The reason is their production costs are lower and their preservative qualities like resistance to light, heat, or pH value. Therefore, synthetic colorings are the most widespread in the industry. The analytic check required to approve a cosmetic product's safety are in place due to their potential secondary effect on human health and the regulation requirements of their usage on the cosmetics market. However, cosmetic ingredients like preservatives, or UV shields can't be identified within cosmetics using the methods available for determining coloring agents. Furthermore, most of the methods at hand only take consideration of a part of regulated coloring [20].

Most of the cosmetic products contain aromatic ingredients in order to make the product more desirable and pleasant for the consumer. Everybody likes a fragrant cosmetic. Aromatics can also be found in hygienic products, perfumes, scented cosmetics, etc. In other words, the usage of said ingredients is quite widespread. Sadly, one of the most notable consequences of coming into contact with aromatics is contact dermatitis. The disadvantage of this healthcare problem is the significant decrease in life quality. Therefore, the security evaluation of cosmetic products must be improved, in order to defend against skin sensitivity more efficiently. In recent decades, chemically induced contact allergy mechanisms were researched extensively, thus giving us a much deeper understanding. There are multiple in vitro methods to identify hazards now [23]. The appearance of chemical allergy is important for toxicologists, who mostly deal with identifying and describing the irritation potential of chemicals on skin (and respiratory system). They also deal with estimating human health risks. The allergic contact dermatitis (ACD) is a health risk that can be avoided entirely in most cases [24]. The safety of cosmetic products can be determined using consecutive steps, which is basically a filtering-type security evaluation.

Based on this, the first question is if the ingredient is used in other body care products. If the answer is "Yes", do we have security guidelines concerning its usage, and in case of "No", tests have to be conducted, like clinical analyses. If we have access to information on safe and secure usage, the question is if we have access to product-specific information on safe and secure usage. If we have no such information, the usage concentration has to be benchmarked to historical usage. In case this is more than the allowed level, there's a need for tests (in silico, in vitro, or clinical). In case the cosmetic product has product-specific information, then we can consider it safe for usage. Unknown, or "No" answers mean that tests have to be conducted (Figure 1) [25]. The physiological function of skin is inversely proportional to the skin's age. These changes are induced by internal (time) and external (mainly UV-induced) factors. Using a variety of plants has a potential advantage in the fight against aging. Such advantages are effects pertaining to anti-inflammation, anti-oxidation, hydration, UV-protection, and other effects. These are mainly found in argan oil, coconut oil, crocin, feverfew, green tea, calendula, pomegranate, and soy [26].

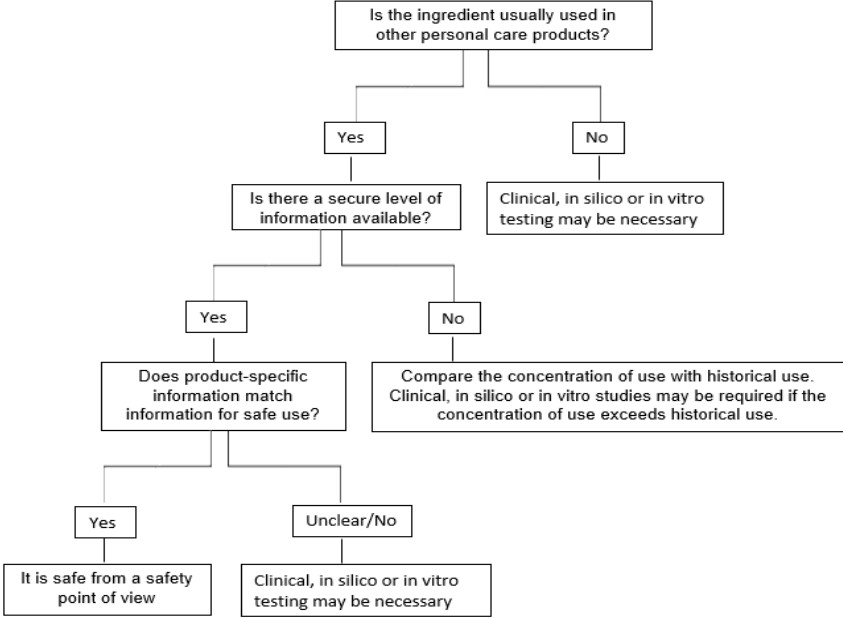

**Figure 1.** Screening level safety assessment framework based on Fung et al. [25].

There is a significant interest towards natural products useful in skincare cosmetics that are non-toxic [27]. Skin whitening products for cosmetic purposes are accessible in commerce. Their function is to make the appearance of skin better [28]. Surface-active ingredients are materials which are a mainstay in the structure of cosmetics, fineries, and personal care products, due to increasing their effects. Dimer surface-active ingredients are better than any of the other, traditional surface-active ingredients in all areas. They are exceptionally promising for usage in a variety of cosmetic products, like shampoos, cremes, and conditioners. Dimers increase the effectiveness of cosmetics to a significant degree, and offer the human epidermis environmentally friendly products to use [29].

*1.3. Environmental Protection and Sustainability on the Cosmetics Market*

Environmental protection problems became a mainstay in public opinion in recent decades. This serves as an incentive for consumers to buy green products. The increasing awareness of consumers towards preferring green products is placing green consumption into focus more and more. Based on the global research of Nielsen in 2015, the number of consumers that wish to reach an increased interest in green products is highest in the Asian-Pacific region [11,30].

In light of this, green strategy became a critical element of business sustainability, since in reality, few companies are capable of realising a green strategy on an organisational level [11,31].

"Green" or "environment-friendly" refer to the values, attitudes, understandings, knowledge, and behavior related to the environment. Companies capable of realising environmental incentives motivate consumers to purchase green products, and also aid the global trend of environmental protection [11].

A reason for social concern in environmental awareness is the depletion of mineral and petrochemical resources, which is also used as reasoning to create a more circular (closed material and energy cycle) economy [32]. In this process, the role of Life Cycle Assessment (LCA) is highly important, since it deals with the analysis of effects products have on the environment, taking the entire production chain, and the product's whole shelf-life into account [33].

Cosmetic and personal body care products are widely used in massive quantities, therefore, their frequent usage cause them to leak back into the environment in similarly massive quantities. There are many products that are biologically active, described by preservative nature and bio-accumulation potential. This means a hazard to the ecosystem and human health. Highly pollutant

materials important from an environmental concern perspective are UV-shields, some preservatives (parabene, triklosane), and plastics [34].

In recent years, industrial process-sourced waste quantity has been increasing steadily. Multiple industries produce a variety of disposable by-products, which are rich in valuable mixtures. Their description and valorisation can not only shape them into highly valuable products for the different areas of bio-technology like cosmetics and pharmaceuticals, but also decrease their effect on the environment, and related treatment costs. Cosmetic active ingredients extracted from fish, meat, and dairy products also have multiple examples. Such are the mixtures and extracts from the waste produced by agricultural or foodstuffs production. Such products are efficient, cheap, and sustainable, thereby offering an alternative to plant-sourced extracts' frequent supply. Furthermore, if waste products are sourced from an eco-farm, they are an even more valuable source of safe extracts needed for these cosmetics, since there are no remaining pesticides, or potentially toxicant materials [35].

Plastic waste management is still a reason for concern for most countries. In the cosmetics industry, some define different strategic alternatives for the same problem's management, which mostly aim at the dedicated planning of packaging [36]. Using heat energy produced from biomass, or renewable energy resources are also viewed as advantageous trends in the production process of cosmetics [37].

*1.4. Safe Cosmetics in the European Union*

The 1223/2009/EK Decree of the European Union rules that companies shall collect and evaluate the reports on the (non-desired) disadvantageous effects of the cosmetics they sell. Apart from this, they also have to report highly problematic non-desired effects to the local authorities. Cosmetics Europe, representing the European cosmetics industry worked out its guideline in order to aid the practice for treating non-desired effects, and reporting serious non-desired effects. Keeping to the guideline makes it possible for the companies in question to show proper care and compliance with legal requirements [38].

*1.5. Responsibility of Cosmetics Producers*

The health awareness of people spread from the foodstuffs industry to the cosmetics industry. Consumers are more and more interested in natural ingredients, sustainable packaging, and other green elements in cosmetics. The cosmetics industry's chemists wage a difficult war against the idea that natural ingredients are safer than their synthetic counterparts. Ingredients need to be selected based on their safety and efficiency, regardless of them being natural or synthetic makeup materials [15]. Cosmetics companies across the globe modified their strategy to handle new challenges better, and incorporate the different aspects of sustainability into their activities. Green marketing highlights a new dimension of economic, social and environmental responsibility taken up by organisations [39]. Cosmetics companies aim more and more to develop their cosmetics in a lab environment, either to be free of chemicals, or to contain as few chemicals as possible. Producers are widening their palette of natural cosmetics to ride the trend of changing consumer attitude [40]. The science of rheology is an efficient tool of cosmetic products, which includes green cosmetics activities, the supply of consumer demands, and the stability of the product (Figure 2) [41]. The original factors of rheology as a powerful tool for cosmetic product design, we have been supplemented based on our analyses. On Figure 2 we are also showing the added factors, like the product likeability (Product flavor), quality (Product quality), and green markings on the product packaging (Green symbols).

This constitutes product symbols which may affect consumer decisions, since a dedicated green consumer will search for these symbols on the packaging. For the green consumer, likeability of the product and the price to performance ratio, and quality are important points to note during consumer decision-making. If the consumer doesn't like the product in question, they won't use the product, however, if there's a gap between the desired and actual product attributes, desired quality cannot be met, in which case the purchase becomes a negative experience for the consumer.

For various industries, nanotechnology serves as the novelty. Nanotechnology is both the newest challenge and newest interest for the cosmetics industry, which they can implement in the manufacture of cosmetics [42]. In the case of greentech solutions, like ultrasound-treatment, organic material remains are used to make the products have a high added value. In general, we can say that plant-sourced extracts can be collected in a variety of ways. Traditional methods usually employ chemical solutions. However, we do know that this also causes a plethora of environmental problems during waste neutralisation [43]. During recent decades, greentech production solutions were of significant interest to modern industries [44]. The more environmental-friendly extraction technology, and the increasing interest of consumers towards natural products caused it to become a potential alternative for the cosmetics industry [45]. Using the extraction technology, decreasing energy usage and pollution, and the quality of extracts improving also had an effect.

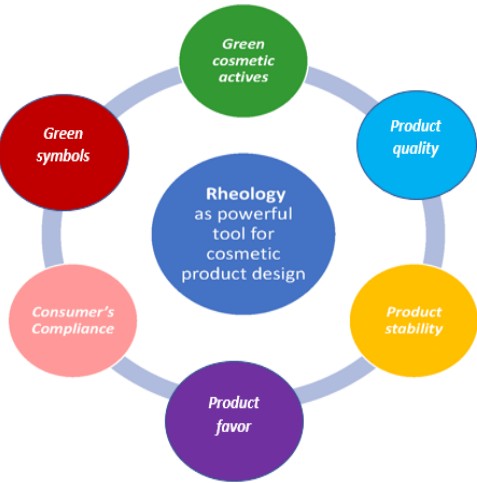

**Figure 2.** Rheology as powerful tool for cosmetic product design (Graphical abstract) based on Semenzato et al. [41].

The zero waste concept inspired the environmentally friendly oil extraction of both natural volatile, and non-volatile bioactive mixtures [46]. The main source of extracting bio-active natural ingredients are plants, most notably herbs and spices [46]. Plant-based active ingredients usable in both traditional pharmaceutical products and functional cosmetics for a longer period, can be applied to the current trend of the green cosmetics market with their well-documented biological effects [41].

Cosmetics demand protection against microbes, just as all water- and organic/inorganic mixture-containing products, to secure the safety of consumers, and increase the shelf-life of the product [47]. The main goal of microbiological safety is the consumer's protection against potentially pathogenic micro-organisms, and avoiding the biological, and physical-chemical degradation of the product. Also important are quality conservation. All of these are supported by chemical, physical, or chemical-physical strategic actions. The most common strategic action is using anti-microbe ingredients. The current validation of the preservative system follows the good manufacture practices, by evaluating the resource and conservation effect using the proper methods [48,49]. UV filters are used in a wide spectrum in cosmetics in order to protect human skin, or the product against the detrimental effects of UV rays. The increasing usage of UV-shields, and their improper placement and storage cause them to be a newfound group among the newly forming pollutants currently in production [50].

*1.6. Green Aspects of Consumer Behavior Related to Cosmetics Purchase*

Consumer behavior underwent a significant change in recent decades, environmental and health awareness obtained a significant role. Consumers also think about future generations, and they consider protecting the state of the environment more and more during their decisions [40]. The trend of using cosmetics gained momentum all around the Globe, however, the actual quantity used is not

determined by any actual statistical data. However, lack of civilian awareness related to cosmetics usage causes a significant healthcare risk, especially in developing countries [51]. According to the theory of planned behavior (TPB), behavioral intention has three factors: Attitudes towards a behavior, subjective norms, and perceived behavior control (PCB) [52].

In 2018, the Cosmetic Toiletry and Perfumery Association (CTPA) summarized that consumer attitudes "use" product attributes, including functions, ingredients, packaging, fragrance materials, and even price, to influence purchase behavior [15]. Green purchase attitudes suggest that the performance of green purchase behavior can be evaluated either positively or negatively. Chan pointed out that environmental awareness and price sensitivity significantly mirrors the level of green purchase attitudes [53]. Knowledge can influence the entire decision-making process of consumers. Knowing green products often comes right before green purchase intent. Green trust and green purchase intent have a connection, positively influenced by perceived price. Higher perceived price results in higher trust from the consumer in the case of green products, therefore, has a higher effect on purchase intent as well [54].

In recent decades, environmental protection and green marketing trends resulted in the change of consumer demand and behavior. Consumers are more and more interested in an environmentally friendly lifestyle, since they not only consider environment protection perspectives, but also want personal advantages from green products. We can see green initiatives on various areas, such as green energy usage during production and manufacture, or the appearance of environmentally friendly, zero waste packaging [55].

Aware consumption prefers natural, and biologically degrading packaging instead of plastic packaging, which in the capacity of their nature as waste, cause environmental damages and load. The cosmetics industry is searching for sustainable solutions in order to increase bio-efficiency and keep the circular economy basics. Their research mainly focuses on naturally and biologically degrading polymers [56]. Increasing environmental awareness in society serves as an incentive for consumers to use green cosmetic products. The fastest growing industry on the global market is the market of green skincare products, as opposed to other green cosmetic products [11].

Consumers love to personalize their cosmetic products, this has become a widespread tendency. They wish to select the ingredients within the products according to the specific needs of their own skin or hair, since they consider personalized products more efficient than the products found on the store shelves [57]. Based on the literature sources, we can assume that the palette of green cosmetics (ingredients, active ingredients, packaging, and technological solutions) is expanding continuously due to research. Legal regulation can also serve as guarantee for safety, and environmental protection is becoming more apparent within the cosmetics industry. Therefore, the main goal of our quantitative research is to introduce the main factors influencing the purchase of green cosmetics, and to determine the current consumer clusters of green cosmetic purchasers in Hungary. Based on the scientific problems and research ideas and goals, the following research questions were formed:

- Are women more prone to buy natural cosmetics than men? Are consumers preferring natural cosmetics more open for newer, more modern natural cosmetics brands during their purchases?
- Are health- and environmentally aware consumers more motivated to buy natural cosmetics in general, compared to the less aware consumers? Are consumers who buy bio-foodstuffs more prone to buy natural cosmetics as well?

## 2. Material and Methods

The questions used in the quantitative analyses were formed by supplementing four questions of the question groups used by Matić and Puh [1] and slight re-wording. Based on their research, the four question groups were the purchase of natural cosmetics, the conditions of buying new natural cosmetics, leading a healthy lifestyle, and consuming bio-foodstuffs as a general eating habit, aiming to offer a positive supplement. Based on the conclusions drawn from the processed literature, in our analyses, the question was modified to assume refusal of purchasing bio-foodstuffs. The questionnaire

was further extended by information on leading an environmentally friendly lifestyle, and we also evaluated the intent to pay extra for a cosmetic product which contains natural ingredients, or the packaging of which was made using natural material. Our intention with the question on refusal of purchasing natural cosmetics was to find out if the purchase intent remains when the product is not as effective as its synthetic counterpart. During the wording of questions, our goal was to make it possible to differentiate between motivation aimed at buying natural or synthetic cosmetics.

### 2.1. Research Material

The data was collected using an online questionnaire in Hungary, which was filled out by a total of 197 participants. We used the resulting purchase pattern in our analyses. The empirical data collection was conducted between 20 April 2018 and 14 May 2018. The questionnaire consisted of two major parts. The first part was a series of seven questions employing the Likert-scale, where participants were asked to define how much they approve of the statements (1—absolutely disapprove, 5—absolutely approve) about their purchase intent towards natural cosmetics, and the statement given ("I wish to buy natural cosmetics."), furthermore, using Yes/No answers. The second part of the questionnaire was a basic collection of the participants' demographic variables (like age and gender).

The questionnaire was filled out by 197 participants, and there are four missing answers. One is missing from the willingness to purchase natural cosmetics, one from environmental awareness, one from packaging, and one from the decision between normal and natural cosmetics. Average values are higher for all variables than the deviation values.

The questionnaire mainly consists of Likert-scales, which means median, minimum, and maximum values were all according to the values of the five level Likert-scale (1–5), and the values based on the choice between two options (1–0, 1–2) (Table 1). Answers between one and three are negative, are basically 'no', whereas are positive for four and five, basically 'yes'. Neutral answers were not included in obvious positive answers, they were rather included in the negative answers, because the answer wasn't completely sure.

**Table 1.** Results of descriptive statistics.

|  | N | Minimum | Maximum | Mean | Std. Deviation |
|---|---|---|---|---|---|
| New brands | 197 | 1.0 | 5.0 | 3.5 | 1.4 |
| Healthy way of life | 197 | 1.0 | 5.0 | 4.1 | 1.0 |
| No biofood' buying | 197 | 1.0 | 5.0 | 2.1 | 1.2 |
| Environmentally conscious consumer behaviour | 196 | 2.0 | 5.0 | 4.3 | 0.7 |
| Natural ingredients | 197 | 1.0 | 5.0 | 4.0 | 1.1 |
| Natural packaging | 196 | 1.0 | 5.0 | 3.9 | 1.2 |
| Less effective natural cosmetics | 196 | 1.0 | 5.0 | 2.4 | 1.4 |
| Valid N (listwise) | 194 |  |  |  |  |

The sampling was done online, the questionnaire was shared using social media via the snowball method. The questionnaire itself could be filled out using the http://ripet.hu online questionnaire site (Appendix B).

### 2.2. Methods

As primary research, we chose to conduct descriptive statistical analyses and cluster analyses using the PASW Statistics 18 programme. First, descriptive statistics were used to analyse the data. As part of the cluster analysis, first, a hierarchic analysis method was chosen (Ward's Cluster), after which a non-hierarchic method (K-means) was used.

(Note: A large number of clustering methods have been proposed for customer segmentation, including hierarchical approaches as well as non-hierarchical approaches. The most popular algorithm of non-hierarchical clustering methods is the K-means algorithm. The K-means algorithm

must pre-specify the number of clusters (K), which is not required for hierarchical algorithms. A non-hierarchical approach would be required to achieve our study goals, but since one of the important conclusions of the research presented by Matić and Puh [1] was that knowledge of the product and its impact is important when making choices, we wanted to see that can be grouped into hierarchical groups. As a result, hierarchical clustering provides an intuitive way to study relationships among clusters not possible using non-hierarchical approaches. Non-hierarchical clustering methods are usually used for larger datasets like the one we had. Based on our hierarchical and non-hierarchic results, we obtained the same clusters, so no further investigations were warranted.)

The results of the analysis were further evaluated using analytical tools, including the methods of analysis and synthesis, inductive and deductive approach method, and the generalisation and specialisation methods. Table 1 clearly shows the 197 participants in the sample, who were willing to fill out the questionnaire, hence, they make up the Hungarian consumer sample of 34% men and 66% women. According to age groups, 17% of participants came from the 18–23, 28% from the 25–34, 23% of the 35–44, 22% of the 45–54, and 11% of the 55 above group (Appendix A).

## 3. Results

### 3.1. Outcome of Descriptive Statistics Analyses

Research results show that 70% of the participants wish to buy natural cosmetics. 56% of them (completely approve and approve categories) wish to buy brands which are newcomers on the cosmetics market. 78% of the participants (completely approve, approve) tend towards following a healthy lifestyle, whereas 68% of participants (completely approve, approve) tend towards buying bio-foodstuffs. 86% of participants (completely approve, approve) are influenced in their decisions by environmental awareness. 70% of the participants are willing to pay extra for a natural cosmetic made of natural ingredients. 68% of participants are similarly willing to pay extra for a cosmetic in packaging made of natural material (completely approve, approve). In terms of a given cosmetic's effectiveness, 57% of participants (completely approve, approve) choose a less effective natural cosmetic instead of a normal cosmetic (Appendix A).

We defined the difference between two participants using the Euclidean distance squared. Data which was further analysed were highlighted using grey in Appendix A and Table 1.

### 3.2. Ward's Cluster Analysis (Hierarchical Analysis)

Variables analysed in the research were: Openness towards new cosmetic brands, health awareness, avoidance of buying bio-foodstuffs, environmental awareness, natural ingredients, natural packaging, and preference towards natural cosmetics opposed to more effective chemical cosmetics despite less effectiveness.

Based on the summary of cases and averages, openness, health awareness, environmental awareness, and natural ingredients and packaging are in Cluster 1. Bio-foodstuffs and buying less effective natural cosmetics were included into Cluster 2 (Table 2).

According to the analysed variables (Table 3), three clusters were outlined. Cluster 1 included (based on the percentage values—above 30%) the purchase of less effective natural cosmetics. Cluster 2 included openness, health awareness, bio-foodstuffs, natural ingredients and packaging, and purchase of less effective natural cosmetics. Cluster 3 included openness, bio-foodstuffs, and the purchase of less effective natural cosmetics. Apart from belonging in the various clusters, the value of final results can also help us make a decision, which means, a difference above 30% is the decisive factor. Cluster 2 includes health awareness and natural ingredients and packaging. Cluster 3 is represented by the openness towards new natural cosmetic brands, bio-foodstuffs, and refusal of purchasing less effective natural cosmetics. Health awareness can't be organized into either cluster.

**Table 2.** A summary of Ward's Cluster analysis.

| Case Summaries | | Ward Method | | | |
|---|---|---|---|---|---|
| | | **1** | **2** | **3** | **Total** |
| New brands | Mean | 4.940 | 2.089 | 3.183 | 3.536 |
| | Std. Deviation | 0.238 | 0.733 | 1.198 | 1.404 |
| Healthy way of life | Mean | 4.955 | 3.000 | 3.915 | 4.062 |
| | Std. Deviation | 0.208 | 0.977 | 0.670 | 0.985 |
| No biofood' buying | Mean | 1.045 | 2.867 | 2.476 | 2.072 |
| | Std. Deviation | 0.208 | 1.099 | 1.135 | 1.189 |
| Environmentally conscious consumer behaviour | Mean | 5.000 | 3.444 | 4.231 | 4.314 |
| | Std. Deviation | 0.000 | 0.624 | 0.551 | 0.747 |
| Natural ingredients | Mean | 5.000 | 2.422 | 3.939 | 3.954 |
| | Std. Deviation | 0.000 | 0.941 | 0.673 | 1.149 |
| Natural packaging | Mean | 4.970 | 2.333 | 3.805 | 3.866 |
| | Std. Deviation | 0.171 | 0.879 | 0.710 | 1.171 |
| Less effective natural cosmetics | Mean | 1.090 | 3.556 | 2.817 | 2.392 |
| | Std. Deviation | 0.379 | 1.078 | 1.167 | 1.366 |

**Table 3.** Deviation/average of variables.

| Variables | Clusters | | |
|---|---|---|---|
| | **1** | **2** | **3** |
| New brands | 4.83% | 35.09% | 37.64% |
| Healthy way of life | 4.21% | 32.57% | 17.13% |
| No biofood' buying | 19.94% | 38.36% | 45.87% |
| Environmentally conscious consumer behaviour | 0.00% | 18.11% | 13.02% |
| Natural ingredients | 0.00% | 38.86% | 17.09% |
| Natural packaging | 3.45% | 37.67% | 18.67% |
| Less effective natural cosmetics | 34.75% | 30.31% | 41.41% |

## 3.3. K-Means Cluster Analysis (Non-Hierarchic Method)

The tri-cluster result is a group close to the main average, Cluster 1 included 71 people, Cluster 2 included 54 people, whereas Cluster 3 included 69 people (Table 4).

**Table 4.** Number of Cases in each Cluster.

| Number of Cases in Each Cluster | | |
|---|---|---|
| Cluster | 1 | 71.0 |
| | 2 | 54.0 |
| | 3 | 69.0 |
| Valid | | 194.0 |
| Missing | | 3.0 |

We can iterate as long as the cluster centres show no more changes (final cluster centres). The three clusters include the main purchase variables (motivation factors) (Table 5).

**Table 5.** The three clusters and Z score variables.

| **Final Cluster Centers** | | | |
| --- | --- | --- | --- |
| **Zsore Variables** | **Cluster** | | |
| | **1** | **2** | **3** |
| Zscore: New brands | 0.956 | −1.092 | −0.153 |
| Zscore: Healthy way of life | 0.862 | −1.063 | −0.067 |
| Zscore: No biofood' buying | −0.851 | 0.662 | 0.372 |
| Zscore: Environmentally conscious consumer behavior | 0.898 | 0.996 | −0.152 |
| Zscore: Natural ingredients | 0.905 | −1.215 | −0.007 |
| Zscore: Natural packaging | 0.918 | −1.191 | −0.027 |
| Zscore: Less effective natural cosmetics | −0.884 | 0.943 | 0.201 |

Figure 3 also supports the tri-cluster solution, which were tagged using three colors. Cluster 1 therefore included openness, health awareness, environment awareness, natural ingredients, and packaging. Cluster 2 included bio-foodstuffs and purchase of less effective natural cosmetics. Cluster 3 included bio-foodstuffs and purchase of less effective natural cosmetics as well. Being sorted into Clusters 2 or 3 was based on the Z score value, according to which, refusal of bio-foodstuff and less effective natural cosmetic purchase means inclusion in Cluster 2, whereas Cluster 3 becomes a hybrid cluster with similar initiative to Cluster 2, however, much less apparent in intent.

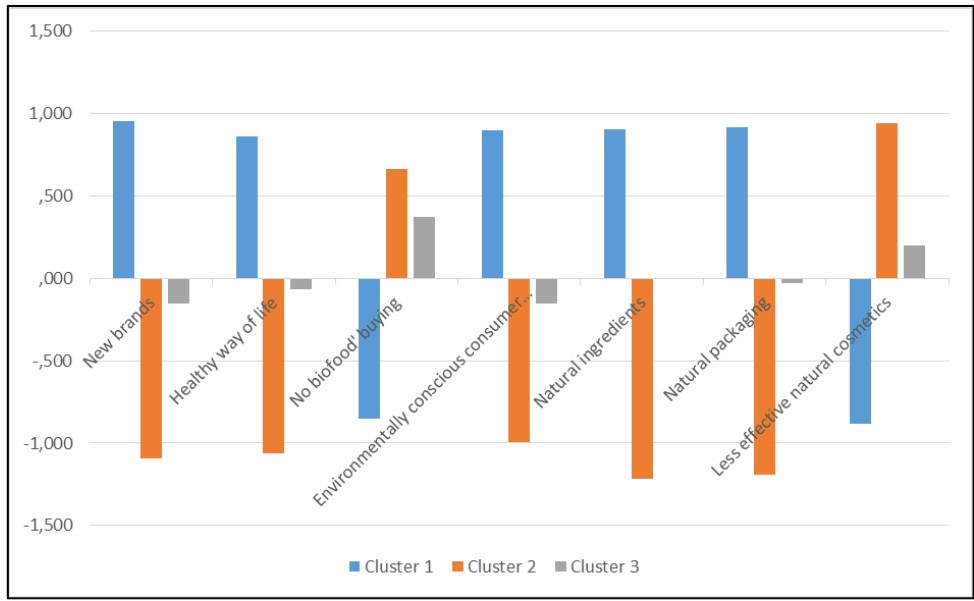

**Figure 3.** Final Cluster Centers with cluster groups.

Cluster 1 can be defined by preferring new brands, health and environmental awareness, and desiring natural ingredients and packaging. Cluster 2 can be defined by avoiding the purchase of bio-foodstuffs, and preferring chemical cosmetics instead of natural cosmetics. Cluster 3 can be defined by a less apparent avoidance of bio-foodstuffs purchase and natural cosmetics, and a slight openness towards natural products. Cluster 1 could be called the NATURAL group, Cluster 2 would be the NON-NATURAL group, whereas Cluster 3 should be named the MIXED group.

As we already discussed in the methodology part, Matić and Puh already conducted a similar analysis for the Croatian consumers, and their results show a similar conclusion to ours. Therefore, the analyses using different methods support the results of each other, on the other hand, in the

case of bio-food consumption versus natural preference, we have the opposite result. For those who prefer natural cosmetics, more people do not consume bio-foods than those who prefer them. The results of the research should be taken into account in the marketing of organic products, because the consumption of bio-food and the trend of the preference of natural cosmetics are very different.

## 4. Discussion

Based on the research conducted, the results showed that three clusters can be formed, based on the consumer behavior on the cosmetics market. One of these is completely green, meaning aims at purchasing natural cosmetics, another one prefers chemical cosmetics, whereas a third cluster is mixed, in which both natural and chemical cosmetics are purchased. These clusters show consumer opinion differences, probably caused by a multitude of factors. There are consumers who buy natural cosmetics, even if those are more expensive than chemical cosmetics, since they consider conserving both their own health and the environment an important factor. Other consumers believe in the traditional, long-time guaranteed brands, but among them are those that are more open for both new products and new brands. The consumer behavior towards cosmetics can be approached in a variety of ways. Based on the results of the primary research, women are more open towards buying cosmetics and natural cosmetics and men, but this is not necessarily in relation to the purchase of bio-foodstuffs. Not all participants found it evident that buying bio-foodstuffs should cause them to prefer natural cosmetics. Consumers won't use natural cosmetics, if they're not as effective as their chemical counterparts. Environmental and health awareness, and preference for natural cosmetics also don't necessarily produce an associative link. Furthermore, different age groups also don't find the importance of natural cosmetics to be the same. Health and environment-aware consumers are more motivated to buy natural cosmetics, but will still choose the intensity of purchase in light of knowledge and information at hand.

Based on the analyses, there are a large variety of preferences when buying natural cosmetics, and to each consumer, different factors may be the defining ones in their final decisions. However, in spite of this, we were able to form three definitive groups using a cluster analysis. These groups have their own main specifics, and factors for consumer decision. Consumer decisions' further analysis should be extended to areas where personal and social/environmental preferences are also taken into consideration for consumer decisions. The apparent and missing knowledge's effect in cosmetics usage should also be analysed in the various target groups and age groups.

Limitations: The validity of research results should only be considered with limitations. The analysis is based on a Hungarian survey, it is advisable to extend the investigations to a larger territorial unit, and to examine them under other income conditions as well. Accurate knowledge of product features, credible information to consumers, and deliberate education of consumers play an important role in decision-making. It is advisable to examine the extent to which education backgrounds, knowledge of a healthy lifestyle, and knowledge of the harmful effects change the consumption habits of each product category. Furthermore, the mechanism of interaction between the products is unknown, which was not investigated in this study. Based on the results of this study, it is not advisable to change the regulatory system or tax conditions.

**Author Contributions:** Conceptualization, C.F.; methodology and formal analysis, N.A.; original draft preparation, N.A.; writing—review and editing with supervision, C.F.

**Funding:** Preparation the manuscript and our final article was supported by the Climate Change Research Centre and Doctoral School of Management and Business Administration at Szent Istvan University.

**Conflicts of Interest:** The authors declare no conflict of interest.

## Appendix A. Sample of the Analysis

**Table A1.** Sample attributes.

| Variables (Questions in the Questionnaire) | Categories | No. of Answers | |
|---|---|---|---|
| | | **Number** | **%** |
| 1. I wish to buy natural cosmetics. | Yes | 137 | 70% |
| | No | 59 | 30% |
| | Total | 196 | 100% |
| 2. I tend to buy new natural cosmetic brands. | Completely disapprove. | 22 | 11% |
| | Disapprove. | 30 | 15% |
| | Neutral/no opinion. | 35 | 18% |
| | Approve. | 38 | 19% |
| | Completely approve. | 72 | 37% |
| | Total | 197 | 100% |
| 3. I have a health-aware lifestyle. | Completely disapprove. | 3 | 2% |
| | Disapprove. | 15 | 8% |
| | Neutral/no opinion. | 26 | 13% |
| | Approve. | 75 | 38% |
| | Completely approve. | 78 | 40% |
| | Total | 197 | 100% |
| 4. I refuse to buy bio-foodstuffs. | Completely disapprove. | 87 | 44% |
| | Disapprove. | 47 | 24% |
| | Neutral/no opinion. | 34 | 17% |
| | Approve. | 21 | 11% |
| | Completely approve. | 8 | 4% |
| | Total | 197 | 100% |
| 5. I'm an environmentally aware consumer, I try not to pollute the environment. | Completely disapprove. | 0 | 0% |
| | Disapprove. | 3 | 2% |
| | Neutral/no opinion. | 24 | 12% |
| | Approve. | 77 | 39% |
| | Completely approve. | 92 | 47% |
| | Total | 196 | 100% |
| 6. I'm willing to pay extra for cosmetics made out of natural ingredients. | Completely disapprove. | 6 | 3% |
| | Disapprove. | 22 | 11% |
| | Neutral/no opinion. | 30 | 15% |
| | Approve. | 54 | 27% |
| | Completely approve. | 85 | 43% |
| | Total | 197 | 100% |

**Table A1.** *Cont.*

| Variables (Questions in the Questionnaire) | Categories | No. of Answers | |
|---|---|---|---|
| | | **Number** | **%** |
| 7. I'm willing to pay extra for cosmetics in natural packaging. | Completely disapprove. | 8 | 4% |
| | Disapprove. | 22 | 11% |
| | Neutral/no opinion. | 33 | 17% |
| | Approve. | 57 | 29% |
| | Completely approve. | 76 | 39% |
| | Total | 196 | 100% |
| 8. I don't buy natural cosmetics less effective than chemical ones. | Completely disapprove. | 77 | 39% |
| | Disapprove. | 36 | 18% |
| | Neutral/no opinion. | 29 | 15% |
| | Approve. | 40 | 20% |
| | Completely approve. | 14 | 7% |
| | Total | 196 | 100% |
| 9. Sex | Male | 67 | 34% |
| | Female | 130 | 66% |
| | Total | 197 | 100% |
| 10. Age | 18-24 | 33 | 17% |
| | 25-34 | 55 | 28% |
| | 35-44 | 45 | 23% |
| | 45-54 | 43 | 22% |
| | 55- | 21 | 11% |
| | Total | 197 | 100% |

## Appendix B. Questionnaire for Online Data Collection

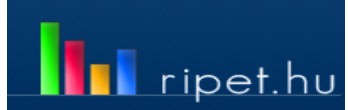

Statistics by question

| 1. I wish to buy natural cosmetics—[196] | | |
|---|---|---|
| | **No.** | **100%** |
| Yes | 137 | 70% |
| No | 59 | 30% |
| **2. I tend to buy new natural cosmetic brands. (1, Completely disapprove, 5, completely approve)—[197]** | | |
| | **No.** | **100%** |
| 1. Completely disapprove. | 22 | 11% |
| 2. Disapprove. | 30 | 15% |
| 3. Neutral/no opinion. | 35 | 18% |
| 4. Approve. | 38 | 19% |
| 5. Completely approve. | 72 | 37% |
| **3. I lead a health-aware lifestyle. (1, Completely disapprove, 5, completely approve)—[197]** | | |
| | **No.** | **100%** |
| 1. Completely disapprove. | 3 | 2% |
| 2. Disapprove. | 15 | 8% |

| 3. Neutral/no opinion. | 26 | 13% |
| 4. Approve. | 75 | 38% |
| 5. Completely approve. | 78 | 40% |

**4. I refuse to buy bio-foodstuffs. (1, Completely disapprove, 5, completely approve)—[197]**

| | **No.** | **100%** |
| --- | --- | --- |
| 1. Completely disapprove. | 87 | 44% |
| 2. Disapprove. | 47 | 24% |
| 3. Neutral/no opinion. | 34 | 17% |
| 4. Approve. | 21 | 11% |
| 5. Completely approve. | 8 | 4% |

**5. I'm an environmentally aware consumer, I try not to pollute the environment (1, Completely disapprove, 5, completely approve)—[196]**

| | **No.** | **100%** |
| --- | --- | --- |
| 1. Completely disapprove. | 0 | 0% |
| 2. Disapprove. | 3 | 2% |
| 3. Neutral/no opinion. | 24 | 12% |
| 4. Approve. | 77 | 39% |
| 5. Completely approve. | 92 | 47% |

**6. I'm willing to pay extra for cosmetics made out of natural ingredients. (1, Completely disapprove, 5, completely approve)—[197]**

| | **No.** | **100%** |
| --- | --- | --- |
| 1. Completely disapprove. | 6 | 3% |
| 2. Disapprove. | 22 | 11% |
| 3. Neutral/no opinion. | 30 | 15% |
| 4. Approve. | 54 | 27% |
| 5. Completely approve. | 85 | 43% |

**7. I'm willing to pay extra for cosmetics in natural packaging. (1, Completely disapprove, 5, completely approve)—[196]**

| | **No.** | **100%** |
| --- | --- | --- |
| 1. Completely disapprove. | 8 | 4% |
| 2. Disapprove. | 22 | 11% |
| 3. Neutral/no opinion. | 33 | 17% |
| 4. Approve. | 57 | 29% |
| 5. Completely approve. | 76 | 39% |

**8. I don't buy natural cosmetics less effective than chemical ones. (1, Completely disapprove, 5, completely approve)—[196]**

| | **No.** | **100%** |
| --- | --- | --- |
| 1. Completely disapprove. | 77 | 39% |
| 2. Disapprove. | 36 | 18% |
| 3. Neutral/no opinion. | 29 | 15% |
| 4. Approve. | 40 | 20% |
| 5. Completely approve. | 14 | 7% |

**9. Sex—[197]**

| | **No.** | **100%** |
| --- | --- | --- |
| Male | 67 | 34% |
| Female | 130 | 66% |

**10. Age—[197]**

| | **No.** | **100%** |
| --- | --- | --- |
| 18–24 | 33 | 17% |
| 25–34 | 55 | 28% |
| 35–44 | 45 | 23% |
| 45–54 | 43 | 22% |
| 55– | 21 | 11% |

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
