# Peer review of "Green Consumer Behavior in the Cosmetics Market"

_resources, doi:10.3390/resources8030137_

Round 1

Reviewer 1 Report

First of all I would like to commend the authors on pursuing a very interesting and timely topic. Indeed, examining the green consumer behaviour is a worthwhile endeavor. However the authors should clearly indicate the objective of their study in the abstract and in the introduction sections. In the conclusion sections the authors should indicate the limitations of their study and the practical implications for cosmetic firms, policy makers and consumers.

Author Response

Dear Reviewer, 

Thank you very much for the valuable suggestions, the suggested corrections, and the improvements were made in coordination with the other reviews. In the case of the abstract and the introductory part, the objective has been defined more specifically. Limitations made to enhance the usability of the results were inserted after the conclusions in the limitation section.

Best wishes, 

Authors

Reviewer 2 Report

Manuscript is interesting, well writting and carefully prepared. The Authors wrote quite long,maybe even too ling introduction, however it contains a lot of information and is written in an exhaustive way. The work does not contain innovative data, but brings new elements to our knowledge about green cosmetics.

Author Response

Dear Reviewer, 

Thank you very much for your criticism and positive feedback. During the presentation of the research background, our aim was to give a clear picture about the consumers of organic food and consumers of natural cosmetics. Sustainability requirements are now a basic requirement for the composition and packaging of products. However, this information is not yet available to consumers in current practice. These will require the development of clear, "best practices" for the future.

Best wishes, 

Authors

Reviewer 3 Report

Natural and green cosmetics are one of the most topical and attractive areas of the modern cosmetic science. This article about the green consumer behaviour is interesting and the experimental part is well developed; however, in my opinion the paper requires a thorough revision of English by a native speaker, because, especially in the introduction, some sentences are idiosyncratic and sometimes their meaning is not totally clear.

Then there are a number of amendments which I would suggest:

- page 6, line 258 and the following: the Authors wrote: “On Figure 2, we also showed...green symbols on packaging.” I do not see any green symbol in Figure 2; please explain.

- page 7, line 296: “UV-filters” is preferable to “UV-shields”.

- page 7 line 309: “aromatics”: the Authors meant fragrances?

I think that this paper deserves to be published after a careful linguistic revision and the suggested fixes.

Author Response

Dear Reviewer, 

Thank you very much for your critical comments and positive feedback. The proposed changes and corrections have been made. The introductory part was reviewed again and sent to a language reviewer. The corrections requested in the review opinion are highlighted in red in the text.

Best wishes, 

the Authors

Reviewer 4 Report

Interesting research paper. But there should be some important points explained: 1. Describe why grouping is needed (i.e., in introduction & literature review); 2. Describe why two methods are used (Ward's linkage and K-means are all clustering methods); and 3. If the clustering analysis is combined with non-parametric analysis, there is always measurement error issue. Authors should indicate how to deal with the measurement error for this combination of methodologies (i.e., clustering + comparison); or 4. Authors may use the advanced level of statistical model for solving the measurement error (e.g., LPA).

Author Response

Dear Reviewer, 

Thank you very much for your critical comments and positive feedback. The proposed changes and corrections have been made. You can find detailed answers to your proposals. 

1.) „Describe why grouping is needed!”

According to the literature and the general opinion, it can be assumed that organic consumers and those who like natural cosmetics belong to the same group, or at least there is a large overlap between the two groups. Preliminary results of the questionnaire survey did not support this assumption, and we, therefore thought that consumers could be classified into groups that had not been studied so far.

2.) Describe why two methods are used (Ward's linkage and K-means are all clustering methods);

A large number of clustering methods have been proposed for customer segmentation, including hierarchical approaches as well as non-hierarchical approaches. The most popular algorithm of non-hierarchical clustering methods is the K-means algorithm. The K-means algorithm must pre-specify the number of clusters (k), which is not required for hierarchical algorithms. A non-hierarchical approach would be required to achieve our study goals, but since one of the important conclusions of the research presented by Matić and Puh (2016) was that knowledge of the product and its impact is important when making choices, we wanted to see that can be grouped into hierarchical groups. As a result, hierarchical clustering provides an intuitive way to study relationships among clusters not possible using non-hierarchical approaches. Non-hierarchical clustering methods are usually used for larger datasets like the one we had. Based on our hierarchical and non-hierarchic results, we obtained the same clusters, so no further investigations were warranted. - an explanation of hierarchical and non-hierarchical methods has been included in the introductory part of Methods, so the explanation of this question can be read in the text, highlighted in red.

3.) If the clustering analysis is combined with non-parametric analysis, there is always measurement error issue. Authors should indicate how to deal with the measurement error for this combination of methodologies (i.e., clustering + comparison) and 4. Authors may use the advanced level of statistical model for solving the measurement error (e.g., LPA).

Thank you for your valuable professional comment and methodological suggestion. The hierarchical analysis was performed for orientation purposes only, and it was not intended to compare the results obtained with different methodologies. The description of the measurement error of the methodologies is not the subject of the research, therefore no special emphasis was put on it.

Based on non-hierarchical analyses, clusters were well defined, so no further analysis was needed. Regarding the purpose of the research, basically, we set orientation goals by exploring the basic relationships.

Best wishes,

the Authors 

Round 2

Reviewer 4 Report

Thank you adjusting the paper by seeing our comments.